# Determinants of incomplete child vaccination among mothers of children aged 12–23 months in Worebabo district, Ethiopia: Unmatched case-control study

**Mesfin Yimer Abegaz** [1] *, **Awol Seid**[1], **Shikur Mohammed Awol**[1], **Seid Legesse Hassen**[2]

1 Department of Public Health St. Paul's Hospital Millennium Medical College, Addis Ababa, Ethiopia,
2 Amhara Public Health Institute, Research and Technology Transfer Directorate, Dessie, Ethiopia

* mesfinyimer13@yahoo.com

**Data Availability Statement:** We have uploaded all relevant data as supplementary information.

## Abstract

In nations like Ethiopia, vaccination rates are low despite being one of the most effective public health treatments to protect infants from common infectious diseases that can be prevented by immunization. In Worebabo District, the reasons of the underutilization of vaccination programs are poorly understood. Therefore, this study aimed to identify determinants of incomplete childhood vaccination in the study setting. Community based unmatched case-control study was carried out among 441 mothers of children aged 12–23 months old (147 cases and 294 controls) in Worebabo District, Ethiopia from March 1—April 30, 2020. Using a multistage sampling process, mothers were chosen. Health professionals were trained to collect data using a pre-tested standardized questionnaire. Data entered into Epi Info version 7.2 and put through statistical analysis in SPSS version 23. Binary logistic regression was performed to determine the odds ratio with a 95%CL. A p-value of under 0.05 was estimated statistically significant. The study found that older moms (>35 years old) were more likely than younger mothers to fail to properly immunize their children (AOR = 2.4, 95% CI: 1.09, 5.28). In addition, mothers with incomplete vaccinations had lower knowledge of the benefits of vaccination (AOR = 2.02, 95% CI: 1.20, 3.39), Negative attitudes towards immunization (AOR = 4.9, 95% CI: 2.82, 8.49), less access to prenatal care (AOR = 3.68, 95% CI: 1.58, 8.54), home delivery (AOR = 5.47, 95% CI: 2.58)., 11.58), absent home visits (AOR = 3.56, 95% CI: 1.69, 7.48), and longer time to reach vaccination site (>1 h) (AOR = 10.07)., 95% CI: 1.75, 57.79) were found associated with mother incomplete vaccination of the child. Mothers being older age, less access to antenatal care services, place of home delivery, longer time to reach vaccination site, negative attitude and poor knowledge towards the benefit of vaccination were associated with mothers' incomplete vaccination of the child. Therefore, health professionals should inform and counsel mothers about the advantages of childhood immunization as well as the consequences of incomplete or not vaccination of children at the time of the facility visit and by community health workers during the routine home visit.

**Funding:** The authors received no specific funding for this work.

**Competing interests:** The authors have declared that no competing interests exist.

## Introduction

Immunization has emerged as the most effective public health measure for the prevention of vaccine-preventable infectious diseases [1–3]. Every year, it saves lives from diseases such as diphtheria, tetanus, pertussis (whooping cough), influenza, and measles [4,5]. Immunization (vaccination) has resulted in the eradication of smallpox, a 94% reduction in the global incidence of polio, and a 94% reduction in neonatal tetanus, as well as dramatic reductions in illness, disability, and death from common childhood diseases [6,7]. Approximately 29% of deaths in children under the age of five are vaccine-preventable worldwide [8,9]. However, 19.9 million infants did not receive the recommended vaccination doses [10]. Furthermore, the patterns of vaccination drop-out rate in Asian and African countries remain consistent from year to year [11]. Ethiopia is one of ten countries where 60% of the world's 19.4 million infants are not immunized routinely [12].

Through the elimination, eradication, and control of life-threatening diseases, vaccination has made a significant contribution to improving public health in the African region [13]. In the last four decades, there has been significant progress in Africa's Expanded Program on Immunization (EPI) [14]. Although overall coverage of routine expanded immunization in Africa has improved, some major challenges remain [14]. Ethiopia, like other African countries, has worked to improve its health system in general, and routine immunization services in particular, in order to reduce the disease burden from vaccine-preventable diseases (VPDs) [15]. Vaccines against ten diseases are currently available in Ethiopia, including measles, diphtheria, Haemophilus influenza type B, tetanus, pertussis, hepatitis B, pneumococcal disease, poliomyelitis, rotavirus infections, and tuberculosis. Vaccines are provided at no cost to all eligible groups before the age of one year [15].

In Ethiopia, the routine immunization program aims to vaccinate more than 3 million infants against 11 antigens each year [16]; however, the 2019 Ethiopian Mini- Demographic Health Survey (mini-DHS 2019) report shows that only 43% of infants are fully vaccinated [17]. Despite the fact that national vaccination program performance is measured in terms of full completion of all vaccines at the recommended doses, full vaccination coverage from all sources demonstrated poor performance [18]. Furthermore, according to the 2019 annual report of the Amhara region's Worebabo district health office, only 71.3% of infants were fully vaccinated, with dropout rates of Penta1 to Penta3 and Penta1 to measles, 6% and 14%, respectively [19]. This dropout rate report was higher than in other districts of the Amhara region's South Wollo zones.

Vaccination coverage is an important indicator of access to and use of immunization services [20–22]; However, it is still tragically underutilized in today's world [23]. Socioeconomic, demographic, and geographic factors all have an impact on childhood vaccination [11]. Although vaccination services are provided free of charge in Ethiopia, coverage of complete vaccination remains low, and the reasons for underutilization of vaccination services in most districts of the Amhara region in general, and Worebabo District in particular, are not well understood and documented. As a result, the purpose of this study was to identify determinants of incomplete child vaccination among mothers of children aged 12–23 months in the Amhara region's Worebabo district, South Wollo zones. Identifying the determinants of incomplete and underutilization of vaccination services is critical for developing effective strategies and overall improvement of vaccination coverage and decreasing the vaccination defaulting rate.

## Methods

### Study design and setting

A community-based unmatched case-control study design was conducted in the Worebabo district of Amhara region, North-East Ethiopia from March 1 to April 30, 2020. The district

has 23 (3 urban and 20 rural) administrative Kebeles (the smallest administrative unit in Ethiopia) [19]. According to the 2007 census, the district had a total population of 120,447 people, with women constituting 50.1% of the population. About 111,307 (92.4%) of the population reside in rural areas. Children under the age of one and children under the age of five also make up approximately 3746 (3.11%) and 16,309 (13.54%) of the total population, respectively. The district has 5 health centers, 20 health posts, and 9 Medium and higher private clinics [19].

## Study population

All mothers of children aged 12 to 23 months old, who had started at least one dose of the routine immunization were used as source population for this study. Among these, two comparison groups (cases and controls) were recruited for the study based on operationally pre-defined outcome variable and selected from the routine immunization registration book at health facility level from January 10 to February 25, 2020. All children aged 12 to 23 months with a history of at least one vaccination were eligible for the study as both cases and controls. Participants of children aged 12 to 23 months that miss at least one from the recommended vaccine and did not complete the vaccination before 1st birthday was as cases. Children in the same age group who had completed all the recommended vaccines (BCG, three doses of pentavalent, three doses of PCV, three doses of OPV, and a measles vaccine) before 1st birthday in selected Kebeles were included as controls.

## Sample size determination and Sampling procedures

The sample size was calculated using two population proportion formulas of STAT CALC of EPI info version 7 with 80% power, 95% confidence interval (CI) with a maximum tolerable error of 5%, and a case to control ratio of one to two (1:2). By taking the above assumptions into account, the maximum sample size was determined based on exposure variables. As a result, the calculated sample size for this study, taking into account the 10% non-response rate and the design effect, was estimated to be (Maximum sample size based on exposure variable + 10% non-response rate) *(design effect) = (266+27) *(1.5) = 441. (147 cases and 294 controls).

The district's study participants were chosen using a multistage sampling technique. First, eight (one urban and seven rural) Kebeles were chosen at random from a total of 23 (three urban and twenty rural) Kebels. The calculated sample size of 441 (147 cases and 294 controls) was distributed proportionally to the size allocation based on the number of cases to each of the selected Kebele (Fig 1). Controls were also assigned based on the case-to-control ratio (one case to two controls) for each Kebele. Then, all incompletely and completely vaccinated children in the selected Kebeles were listed from the routine immunization registration book as a sampling frame. The study participants were then chosen using a systematic random sampling technique from each Kebele's sampling frame. The primary caregivers of the chosen children were contacted in order for them to participate in the study.

## Data collection

Data were collected using a structured questionnaire adapted from the previous study [24]. The English version of the questionnaire was translated into Amharic (the local language) for data collection and then back into English to ensure consistency. Six nurse professionals (two diploma and four BSc nurses) collected data from mothers/caretakers directly involved in child care, who were supervised by two district health office supervisors. Furthermore, the vaccination status of the child was confirmed using the vaccination card and the mothers'/caretakers' oral (verbal) responses.

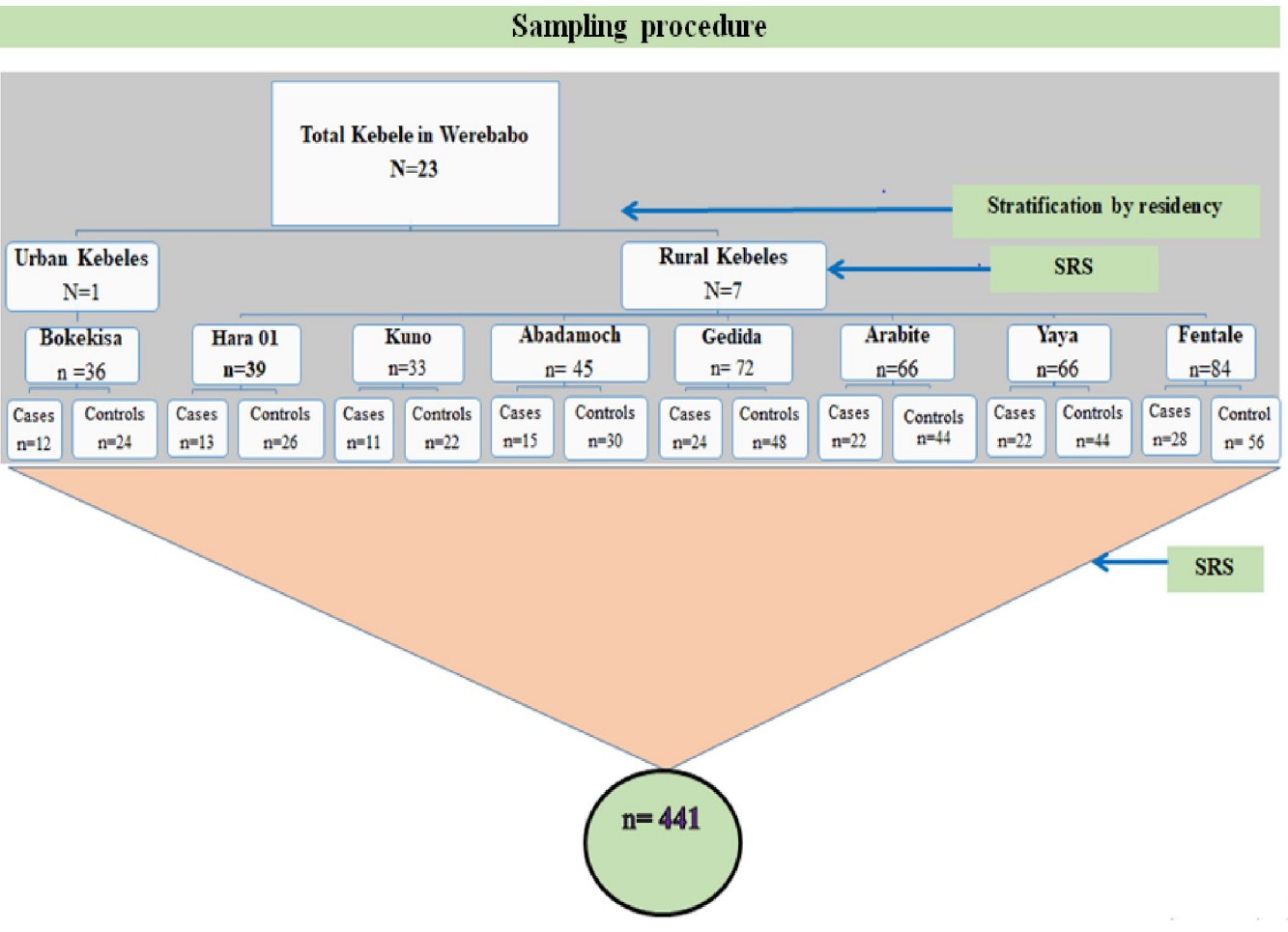

**Fig 1. Schematic illustration of the sampling procedure of study participants in Worebabo district, 2020.**

## Data processing and analysis

All data collectors and supervisors received one-day training on the purpose of the study, study participant selection, data collection Kebeles, data collection tool, and collection techniques to minimize measurement bias. To identify potential problems encountered during data collection, 5% of the actual study subjects (10 from Bistima 01 Kebele and 14 from Gedida Kebele) were pretested outside of the study area. Data were entered into Epi-info 7.2.1 software and exported to SPSS statistical software version 23 for cleaning, recoding, and analysis. Frequencies and cross-tabulation of exposure variables with the main outcome variable were used to clean the data. Before proceeding with the analysis, the missing values were checked. During the analysis, the frequencies and summary measures of the various variables, such as percentage and mean, were determined.

Then, using simple and multiple binary logistic regression analysis models, inferential statistics were performed to measure the strength and direction of associations between the exposure and outcome variables and to identify predictors of childhood incomplete vaccination. To assess the effect of a confounder, exposure variables with a p-value of 0.25 during simple binary logistic regression analysis were included in the final model (multiple binary logistic regression analysis). A backward stepwise logistic regression method was used in the multiple

binary logistic regression analysis. To determine the significant association of predictor variables with the vaccination status of the child, a P-value of 0.05 and a 95% confidence interval for each crude (COR) and adjusted odds ratio (AOR) value were used, and results were presented in texts, tables, and graphs.

## Operational definitions

### Complete vaccination

A child was considered completely vaccinated if he or she received all of the recommended basic vaccines, including one dose of BCG, three doses of DPT-HepBHib (pentavalent), three doses of polio vaccines, three doses of PCV, and one dose of measles vaccine before their first birthday.

### Incomplete vaccination

Participants were classified as having incomplete immunization when they received only some of the vaccines and/or did not receive the full dose of all vaccines before their first birthday.

### Perception

Using two Likert Scale questions, we assessed people's attitudes toward health-care institutions and their views on vaccine side effects. The mean score for each construct was calculated 4.02 mean score for perception for health institution support and 3.42 mean score for perception for vaccine side effect and finally classified as positive or negative. Those who scored lower than the mean ($\leq 4.02$ and $\leq 3.42$) respectively was labeled as having negative perceptions.

### Maternal/Caretaker knowledge

Six immunization knowledge-related questions were used to assess maternal/caretaker knowledge on child immunization. Then correct answers received a one, while incorrect answers received a zero. Those who scored higher than the mean score ($>5.25$) was thought to have good knowledge, while those who scored lower than the mean score ($\leq 5.25$) was thought to have poor knowledge.

### Attitude toward immunization service

Three five-point Likert scale immunization-related variables were used to assess mothers' attitudes toward immunization service. The mean was then computed using three variables. Positive attitudes were defined as scores greater than the mean score ($>10.9$), while negative attitudes were defined as scores less than the mean ($\leq 10.9$).

### Ethical approval and consent to participate

This study was performed after obtaining ethical approval from the Research Ethics Committee of St. Paul's Hospital Millennium Medical College (SPHMMC). Letters of support were also received from the head of the health department and health center in Worebabo district. We got an informed written informed consent from each participant, which was approved by the ethical review committee. The interviewers took precautionary measures a brief discussion and explanation with the mothers or caregivers about the purpose, benefits and anticipated risks of the study having children aged 12–23 Months before the interview. The study was conducted in accordance with the Declaration of Helsinki.

## Result

### Socio-demographic characteristics of mothers/caregivers and children

A total of 441 (147 cases and 294 controls) of mothers/caretakers of children aged between 12–23 months from eight Kebeles participated in this study with 100% response rate. Among the total respondents, 405 (91.8%) were from rural areas and 226 (51.2%) of the children were female. The largest proportion 141 (95.9%) of cases and 291 (98.9%) of controls were biological mothers served as an immediate caretaker for their children.

The mean ± standard deviation ages of respondents (mothers/caretakers) and children in both groups (cases and controls) were 29±5.83 years and 17.4±3.1 months respectively. Besides, the mean ± standard deviation ages of mothers/caretakers among cases and controls were 30± 6.93 and 28.3± 5.07 respectively. About 212 (48.1%) of mothers in both groups (cases and controls) were in the age group 27–34 years. Concerning the marital status of mothers/caretakers, the majority 416 (94.3%) mothers/caretakers were married. While, the rest 25 (5.7%) were either never married, widowed, or separated (not in a union). Nearly half 200 (45.4%) of mothers/caregivers of the children in both study groups; 89 (60.5%) cases and 111 (37.8%) controls had unable to read and write by educational level.

The average family size of the study participants was 5, in which almost half 223 (50.6%) of the families had greater and above 5 members. Also, half 224 (50.8%) of the mothers had greater and above three children, and among them, 79 (53.7%) were from incompletely vaccinated children. The majority of 209 (47.4%) of respondents were in the birth order category between 2nd and 3rd birth orders which comprises 66 (44.9%) of cases and 143 (48.6%) of controls. The majority 301 (68.3%) of the children in both groups (cases and controls) had 21 to 47 months' birth interval between the index child and the preceding child (Table 1).

Out of 441 respondents, 253 (57.4%) mothers/caretakers scored above the mean ± SD score value of 5.25±1.21 from the computed knowledge-related variables and they were categorized as with good knowledge. Regarding the attitude of mothers or caretakers towards immunization, most 290 (65.8%) of mothers in both groups scored above the mean ± SD score value of 10.9 ±1.62 and categorized as with a positive attitude towards immunization. The mean ± SD score value of mothers or caretakers on their perception towards vaccines side effect was 3.42 ±1.1 and 249 (56.5%) respondents from both groups were considered and categorized as with positive perception towards vaccines side effect. The other measure of perception of mothers or caretakers of the children was perception towards health institute support. Consequently, the mean ± SD score value of respondents on the perception towards health institution support was 4.02±1.0, and 312 (70.7%) of the mothers or caretakers from both groups had a negative perception.

### Health service-related characteristics of mothers/caretakers

Regarding ANC visits in maternal health service of the total respondents of both groups (cases and controls), 391 (88.6%) mothers had ANC visits during their pregnancy period. Besides, of the total ANC visited mothers from both groups (cases and controls); about 49 (12.5%) mothers had within two ($\leq$ 2) total ANC visits, and 19 (38.7%) were from incompletely vaccinate children.

Almost fourth-fifths of the respondents, 374 (84.8%) from both groups (cases and controls) preferred to deliver their child at the health institute (health centers/health posts). Of those delivered in health institutes, the majority of 365 (97.5%) of both groups were advised to vaccinate their child. Of those mothers or caretakers who had gotten advice about child vaccination, 94 (25.7%) mothers were incompletely vaccinated children. More than half of primary

**Table 1. Socio-demographic characteristics of mothers/caregivers and children in Worebabo district, Amhara region, Ethiopia, 2020.**

| Variables | Category | Cases, n = 147 | Controls, n = 294 |
|---|---|---|---|
| Immediate caretaker | Mother | 141 (95.9%) | 291(99%) |
| | Father or other relatives | 6 (4.1%) | 3 (1%) |
| Religion | Muslim | 143 (97.3%) | 279 (94.9%) |
| | Orthodox | 4 (2.7%) | 15 (5.1%) |
| Living in couple | Yes | 133 (90.5%) | 283 (96.3%) |
| | No | 14 (9.5%) | 11 (3.7%) |
| Ethnicity | Amhara | 145 (98.6) | 289 (98.3%) |
| | Others | 2 (1.4%) | 5 (1.7%) |
| Occupation (Mother) | House wife | 141 (95.9%) | 280 (95.2) |
| | Student | 3 (2%) | 2 (0.7%) |
| | Others | 3 (2%) | 12 (4.1%) |
| Educational status (Mothers/caretakers) | Unable to read and write | 89 (60.5%) | 111 (37.8%) |
| | Read and write | 24 (16.3%) | 78 (26.5%) |
| | Elementary | 29 (19.7%) | 87 (29.6%) |
| | Secondary and above | 5 (3.4%) | 18 (6.1%) |
| Family Size | <5 Families | 66 (44.9%) | 152 (51.7%) |
| | ≥5 Families | 81 (55.1%) | 142 (48.3%) |
| Parity (Number of children) | 1 | 29 (19.7%) | 71 (24.1%) |
| | 2 | 39 (26.5%) | 78 (26.5%) |
| | ≥3 | 79 (53.7%) | 145(49.3%) |
| Age (Child) | 12–18 Months | 83 (56.5%) | 180 (61.2%) |
| | 19 – 23Months | 64 (43.5%) | 114 (38.8%) |

**Others for ethnicity**: Oromo and Tigre; **Others for occupation**: Farmer, daily laborer, merchant, Government employed; **Living in a couple**: Married labeled as "Yes" and Single, divorced, widowed and separated labeled as "No".

caretakers, 91 (61.9%) of cases, and 106 (36.1%) control had within two (≤2) postnatal visits. About 119 (81%) of mothers of the cases and 278 (94.6%) controls who attended postnatal care (PNC) were also advised to vaccinate their child during their PNC visits.

The availability and accessibility of vaccination services were assessed by the presence of the service and average walking time to the health facility. As a result, the mean ± SD score value of distance to the vaccination site is 8.1±5.2 Km. Moreover, the mean ± SD score value of walking time to reach the vaccination site takes 68.66 ±48.3 minutes. For the majority of respondents, 408 (92.5%) among both groups (cases and controls) were forced to go more than 2 km to get the vaccination.

## Maternal, child, and family-related characteristics association with childhood vaccination

Risk factors for childhood incomplete vaccination were further identified using a case-control study conducted among 441 study participants. Then, twenty-nine variables were included in simple binary logistics regression for statistical analysis. Concerning the socio-demographic and family-related characteristics of study participants found that; being unable to read and write of the education level, older age of mothers or caretakers, living in a couple of marital status and children had care by their fathers or other relatives were significantly associated with childhood incomplete vaccination (Table 2).

**Table 2. Maternal, child, and family-related characteristics association with childhood vaccination in Worebabo district, 2020 (n = 441; cases: 147 and controls: 294).**

| Variables | Category | Vaccination status | | Bi -variable analysis | | |
|---|---|---|---|---|---|---|
| | | Cases | Controls | COR | 95% CI | P-Value |
| Age of mothers/caretakers | ≤ 26 years | 40 (27.2%) | 117 (39.8%) | 1 | | |
| | 27–34 years | 74 (50.3%) | 138 (46.9%) | 1.56 | 0.99–2.47 | 0.053 |
| | ≥ 35 years | 33 (22.4%) | 39 (13.3%) | 2.47 | 1.37–4.44 | 0.002* |
| Religion | Muslim | 143 (97.3%) | 279 (94.9%) | 1.92 | 0.62–5.89 | 0.253 |
| | Orthodox | 4 (2.7%) | 15 (5.1%) | 1 | | |
| Living in couple | Yes | 133 (90.5%) | 283 (96.3%) | 1 | | |
| | No | 14 (9.5%) | 11 (3.7%) | 2.70 | 1.19–6.12 | 0.017* |
| Ethnicity | Amhara | 145 (98.6) | 289 (98.3) | 1.25 | 0.24–6.54 | 0.788 |
| | Others | 2 (1.4%) | 5 (1.7%) | 1 | | |
| Occupation (Mothers/caretakers) | Housewife | 141 (95.9%) | 280 (95.2) | 1.17 | 0.44–3.12 | 0.746 |
| | Others | 6 (4.1%) | 14 (4.8%) | 1 | | |
| Education level (mother/caretaker) | Unable to read and write | 89 (60.5%) | 111 (37.8%) | 2.88 | 1.03–8.08 | 0.044* |
| | Read and write | 24 (16.3%) | 78 (26.5%) | 1.1 | 0.37–3.29 | 0.854 |
| | Elementary | 29 (19.7%) | 87 (29.6%) | 1.2 | 0.40–3.52 | 0.74 |
| | Secondary and above | 5 ((3.4%) | 18 (6.1%) | 1 | | |
| Residential place | Urban | 12 (8.2%) | 24 (8.2%) | 1 | | |
| | Rural | 135 (91.8%) | 270 (91.8%) | 1.00 | 0.48–2.06 | 1.000 |
| Immediate caretaker | Mother | 141 (95.9%) | 291(99%) | 1 | | |
| | Father or other relatives | 6 (4.1%) | 3 (1%) | 4.12 | 1.01–16.74 | 0.047* |
| Family Size | <5 Families | 66 (44.9%) | 152 (51.7%) | 1 | | |
| | ≥5 Families | 81 (55.1%) | 142 (48.3%) | 1.31 | 0.88–1.95 | 0.178 |
| Parity (Number of children) | 1 | 29 (19.7%) | 71 (24.1%) | 1 | | |
| | 2 | 39 (26.5%) | 78 (26.5%) | 1.22 | 0.68–2.19 | 0.493 |
| | ≥3 | 79 (53.7%) | 145(49.3%) | 1.33 | 0.80–2.22 | 0.270 |
| Age (Child) | 12–18 Months | 83 (56.5%) | 180 (61.2%) | 0.82 | 0.55–1.22 | 0.337 |
| | 19 – 23Months | 64 (43.5%) | 114 (38.8%) | 1 | | |
| Sex (Child) | Male | 71 (48.3%) | 144 (49.0%) | 1 | | |
| | Female | 76 (51.7%) | 150 (51.0%) | 1.02 | 0.69–1.52 | 0.893 |
| Birth Order (Child) | 1st | 30 (20.4%) | 69 (23.5%) | 0.69 | 0.4–1.21 | 0.205 |
| | 2nd– 3rd | 66 (44.9%) | 143 (48.6%) | 0.74 | 0.47–1.17 | 0.199 |
| | 4th above | 51 (34.7%) | 82 (27.9%) | 1 | | |
| Birth interval (Child) | ≤23 Months | 8 (6.8%) | 22 (9.9%) | 1 | | |
| | 21–47 Months | 106 (89.8%) | 195 (87.8%) | 1.49 | 0.64–3.47 | 0.350 |
| | ≥48 Months | 4 (3.4%) | 5 (2.3%) | 2.2 | 0.47–10.3 | 0.317 |

**Others for ethnicity**: Oromo and Tigre; **Others for occupation**: Farmer, daily laborer, merchant, Government employed; **Living in a couple**: Married labeled as "Yes" and Single, divorced, widowed and separated labeled as "No".

## Mothers' knowledge, perception, and attitude and health services

Mothers who had poor knowledge of immunization, mothers with a **negative** perception towards health institution support, and with a negative attitude of immunization were significantly associated with childhood incomplete vaccination (Table 3). Regarding maternal and child health care services, mothers who did not use ANC visits during their pregnancy of the index child, home delivery of the child, and lacked PNC follow-up were significantly associated with childhood incomplete vaccination (Table 3).

**Table 3. Mothers knowledge, perception, and attitude and health services related characteristics association with childhood vaccination in Worebabo district, 2020 (n = 441; cases: 147 and controls: 294.**

| Variable | Category | Vaccination status | | Bi variable analysis | | |
|---|---|---|---|---|---|---|
| | | Cases | Controls | COR | 95% CI | P-Value |
| Knowledge towards immunization | Good | 60 (40.8%) | 193 (65.6%) | 1 | | |
| | Poor | 87 (59.2%) | 101 (34.4%) | 2.77 | 1.84–4.16 | <0.001* |
| Perception towards side effect | Positive | 77 (52.4%) | 172 (58.5%) | 1 | | |
| | Negative | 70 (47.6%) | 122 (41.5%) | 1.28 | 0.86–1.09 | 0.222 |
| Perception on health institution support | Positive | 27 (18.4%) | 102 (34.7%) | 1 | | |
| | Negative | 120 (81.6%) | 192 (65.3%) | 2.36 | 1.45–3.82 | <0.001* |
| Attitude towards immunization service | Positive | 67 (45.6%) | 223 (75.9%) | 1 | | |
| | Negative | 80 (54.4%) | 71 (24.1%) | 3.7 | 2.46–5.71 | <0.001* |
| ANC visit | Yes | 109 (74.1%) | 282 (95.9%) | 1 | | |
| | No | 38 (25.9%) | 12 (4.1%) | 8.1 | 4.12–16.26 | <0.001* |
| Delivery | Health facility | 98 (66.7%) | 276 (93.9%) | 1 | | |
| | Home | 49 (33.3%) | 18 (6.1%) | 7.66 | 4.26–13.79 | <0.001* |
| Health service (Child) | Yes | 100 (68%) | 212 (72.1%) | 1 | | |
| | No | 47 (32%) | 82(27.9%) | 1.21 | 0.79–1.86 | 0.375 |
| Having immunization Card | Yes | 106 (72.1) | 226 (76.9%) | 1 | | |
| | No | 41 (27.9%) | 68 (23.1%) | 1.28 | 0.81–2.01 | 0.275 |
| Long waiting line during the last vaccination | No | 142 (99.6%) | 278 (94.6%) | 1 | | |
| | Yes | 5 (3.4%) | 16 (5.4%) | 3.4 | 0.58–4.55 | 0.347 |
| PNC visit | Yes | 122 (83%) | 279 (94.9%) | 1 | | |
| | No | 25 (17%) | 15 (5.1%) | 3.8 | 1.94–7.48 | <0.001* |
| Informed next vaccination date | Yes | 131 (89.7%) | 285 (96.9%) | 1 | | |
| | No | 16 (10.9%) | 9 (3.1%) | 3.86 | 1.66–8.98 | 0.002** |
| Monthly visited by health professionals | Yes | 112 (76.2%) | 272 (92.5%) | 1 | | |
| | No | 35 (23.8%) | 22 (7.5%) | 3.86 | 2.17–6.87 | <0.001* |
| Returned without vaccination during the last vaccination date | No | 122 (83%) | 251 (85.4%) | 1 | | |
| | Yes | 25 (17%) | 43 (14.6%) | 1.19 | 0.69–2.04 | 0.514 |
| Distance from the Vaccination site | ≤ 2 Km | 9 (6.1% | 24 (8.2%) | 1 | | |
| | > 2 Km | 138 (93.9%) | 270 (91.8%) | 1.36 | 0.61–3.01 | 0.444 |
| Time taken to reach the vaccination site | < 15 min | 3 (2%) | 23 (7.8%) | 1 | | |
| | 15–30 min | 20 (13.6%) | 47 (16%) | 3.26 | 0.87–12.11 | 0.077 |
| | 31–60 min | 51 (34.7) | 114 (38.8% | 3.43 | 098–11.94 | 0.053 |
| | > 60 min | 73 (49.7%) | 110 (37.4%) | 5.08 | 1.47–17.56 | 0.010* |

ANC: Antenatal care service; PNC: Postnatal Care service.

## Determinants of incomplete child vaccination

Comparison of variables that were statistically significant with the child immunization status on the crude analysis was adjusted for possible confounders. Sixteen variables [primary caretaker for the child, age of mothers, marital status, education level, family size, birth order, knowledge of mothers, perception of mothers towards vaccine side effect, primary caretaker's (mother's) perception towards health institution support, mother's attitude towards immunization, ANC, delivery, PNC, informed about the next immunization date, health worker home visits and time take to reach vaccination site] were included in a multivariable logistic regression analysis model.

**Table 4. Independent predictors associated with incomplete childhood vaccination in Worebabo district, Ethiopia, 2020 (n = 441; cases: 147 and controls: 294).**

| Variable | Category | Multi-Variable analysis | | |
|---|---|---|---|---|
| | | AOR | 95% CI | P-Value |
| Age (Mothers/caretakers) | ≤ 26 Years | 1 | | |
| | 27–34 Years | 1.39 | 0.79–2.45 | 0.252 |
| | ≥35 Years | **2.40** | **1.09–5.28** | **0.028*** |
| Education Level (Mothers/caretakers) | Unable to read and write | 2.84 | 0.81–9.91 | 0.101 |
| | Read and write | 1.32 | 0.36–4.79 | 0.666 |
| | Elementary | 1.18 | 0.32–4.30 | 0.794 |
| | Secondary and above | 1 | | |
| Knowledge on immunization (Mothers/Caretakers) | Good | 1 | | |
| | Poor | **2.02** | **1.20–3.39** | **0.007*** |
| Attitude on immunization service (Mothers/Caretakers) | Positive | 1 | | |
| | Negative | **4.9** | **2.82–8.49** | **< 0.001*** |
| ANC Visit | Yes | 1 | | |
| | No | **3.68** | **1.58–8.54** | **0.002*** |
| Delivery place | Health facility | 1 | | |
| | Home | **5.47** | **2.58–11.58** | **<0.001*** |
| PNC Visit | Yes | 1 | | |
| | No Visit | 2.19 | 0.88–5.45 | 0.091 |
| Health professional monthly household visit | Yes | 1 | | |
| | No | **3.56** | **1.69–7.48** | **0.001*** |
| Time taken to reach the vaccination site | < 15 min | 1 | | |
| | 15–30 min | 4.74 | 0.78–28.55 | 0.089 |
| | 31–60 min | 4.57 | 0.82–26.06 | 0.087 |
| | > 60 min | **10.07** | **1.75–57.79** | **0.010*** |

ANC: Antenatal care service; PNC: Postnatal Care service.

Thus, the multivariable logistic regression analysis result showed that seven variables were independent predictors for the incomplete childhood vaccination (Table 4). Consequently, children born from older-aged mothers (≥35 years) were 2.4 times more likely to be incompletely vaccinated (AOR = 2.4, 95% CI: 1.09, 5.28) as compared to younger mothers aged (≤26 years). Moreover, the study revealed that children's exposure to incomplete vaccination was increased when primary caretakers of the child who had poor knowledge of immunization had higher odds of being incompletely vaccinated (AOR = 2.02, 95% CI: 1.20, 3.39) as compared to counterparts.

Children who were born from mothers who had no ANC follow-up during their pregnancy were 3.7 times more likely to remain incomplete their vaccination as compared to mothers/caretakers who had ANC visits (AOR = 3.7, 95% CI: 1.85, 8.54). Another finding, mothers or caretakers who were not visited (lacked monthly household visit) by health professionals within one year were 3.6 times more likely to have incompletely vaccinated children (AOR = 3.56, 95% CI: 1.69, 7.48) as compared the counterparts. Concerning the vaccination service availability and accessibility, the distance of the vaccination site was assessed by the presence of the service and average walking time to reach the vaccination sites. Accordingly, mothers or caretakers were living far and the time taken to reach the vaccination sites above one hour had ten times more likely to have incompletely vaccinated children (AOR = 10.07, 95% CI: 1.07, 57.79) compared to travel < 15 minutes.

## Discussion

Immunization is one of the most powerful and cost-effective of all health interventions. Factors determining incompletion of vaccination are complex. Our study determined the predictors of incomplete vaccination among children in the Worebabo district. The overall finding of the final adjusted logistic regression analysis result indicated that age of the mothers/caretakers, antenatal care (ANC) service during pregnancy, place of index child delivery, health professional monthly household visit, knowledge of mothers/caretakers towards immunization, the attitude of mothers or caretakers towards immunization and time taken to reach the vaccination sites were the independent predictors of incomplete childhood vaccination.

Among socio-demographic characteristics of respondents identified during analysis, the age of mothers/caretakers was found to be the only independent factor that leads to incomplete vaccination of the child. Thus, children born and cared for by older mothers were 2.4 times more likely to have incompletely vaccinated children as compared to younger mothers/caretakers (≤26 years). Our finding is in agreement with previous cross-sectional studies conducted in Jigjiga, Ethiopia, and Ghana in 2016 [25,26]. It is well recognized that the age of mothers plays an important role in the utilization of maternal and child health services including immunization. This may be younger women are more aware and active than older mothers to accept and use the maternal and child health services including childhood immunization. In the contrary, other similar case-control studies conducted in Kenya and the Arbagona district of South Ethiopia show that children born from younger mothers (< 25 years) and (≤19 years) were found to be more likely incompletely vaccinated [24,27]. This may be due to older mothers may have a good awareness about the benefits of child health services and may give due attention to getting full immunization of their children more than younger mothers.

Among those maternal health services-related factors; mother's antenatal care (ANC) service, place of childbirth, and a monthly visit by health professionals were found to be independent predictors that lead to incomplete vaccination of the child. Consequently, mothers' use of antenatal care services during their pregnancy was one of the variables significantly associated with the vaccination status of the child. The result indicates that children who were born from mothers who had no antenatal care visit during their pregnancy period were 3.7 times more likely to default to complete vaccination compared to infants who were born from mothers who had antenatal care visits. This finding is consistent with the similar studies conducted in Machakel District and South and South West of Ethiopia which show similar findings to default their vaccination [28–31]. This could be due to mothers who had a visit to antenatal and postnatal care services have a better chance of communicating with health care workers and receiving comprehensive maternal and child health services with health education and counseling about the benefit, schedule, and side effect of the vaccination. Hence, the impact of prenatal and postnatal care information mainly operates through the role of health providers in the process of providing the right and up-to-date information to mothers. Birth preparedness and immunization counseling are important components of prenatal care.

Similarly, this study revealed that home delivery remained a strong predictor for incomplete vaccination. Children who were born at home were 5.47 times more likely to incompletely vaccinate compared to children born at a health institution. The finding of our study is similar to the other studies conducted in Nepal, Machakel District, South and South-west Ethiopia, and Pakistan [26,28,29,31–33]. This could be explained by mothers who give births at health institutions have better access to health education, counseling, and child health services including vaccination of the child. Moreover, most of the time the first dose of the vaccination is given just after birth in the health facility.

Maternal knowledge status positively influenced the completion of childhood immunization in our study. Thus, children whose mothers/caretakers had poor knowledge of immunization were 2.02 times more likely remained to have incompletely vaccinated children than mothers who had good knowledge. This finding is similar to the recent studies carried out in Hawassa Zuria, Laelay Adiabo districts, Gondar City administration of Ethiopia, and South Ethiopia [29,34–36]. This shows that knowledge of a mother on a particular or a given health service including immunization creates a conducive situation for the utilization of the service. Besides, the possible explanation could be that if the primary caretakers have good knowledge about the benefit, schedule, and side reaction of the vaccination, they could immunize their children based on the recommended schedule without defaulting and fearing the side reaction due to vaccination.

Moreover, the attitude of mothers or immediate caretakers towards immunization was found as an important determinant of childhood immunization in our study. Mothers or caretakers who had a negative attitude about immunization were 4.9 times more likely to have incompletely vaccinated children than mothers or immediate caretakers who had a positive attitude. A similar positive association finding was obtained from another cross-sectional survey which was done in Dschang, West Region, Cameroon [37].

Another finding which shows a positive association with incomplete vaccination among the maternal and child health care services related variables was health professional's household visit. Thus, mothers/caretakers who had no household visits by health professionals within a year were 3.6 times more likely to default to complete their vaccination compared to mothers/caretakers who had visited. This finding is in line with the study done in the Laelay Adiabo and Jigjiga districts of Ethiopia [26,34]. This indicates that health professional household visits especially by health extension workers in Ethiopian settings are an important medium for dissemination of all health-related information via health education including immunization services. This can also provide an opportunity in tracing children who default their vaccination. Household visits contributed to higher rates of immunization, especially in the case of illiterate mothers, delivered at home and mothers who had no health education on immunization [38].

Time taken to reach the vaccination sites was an independent predictor of defaulting from completion of child immunization. Mothers or caretakers who need more than one hour of traveling time to reach the vaccination sites were 10 times more likely to have incompletely vaccinated children than those who have within 15 minutes traveling time. The finding is coherent with the previous and similar study done in the Laelay Adiabo district of Ethiopia [34,35]. Our finding is also supported by the study conducted in Minjar Shenkora, Ethiopia in 2017, mothers who traveled on foot for greater than two hours more were likely to incomplete vaccination [39]. This indicates that mothers traveled a long distance and spent more time on the travel to arrive at their vaccination sites might potentially affect their day-to-day activities. This probably may lead mothers to prefer their routine activities to travel a long distance to the immunization site for vaccinating their child.

In the socio-demographic variable, the age of mothers/caretakers was the only significantly associated factor leads to incomplete childhood vaccination. But, different studies have shown that the educational status of mothers/caretakers (Unable to read and write) was found as an important factor leading to incomplete vaccination of the child as compared to educated mothers (secondary and above) [29,40]. The finding of our study may be due to socio-cultural and attitudinal variation among study participants in different study areas. Moreover, in contrary to other studies that were conducted in different areas remaining predictor variables such as birth order of the child, postnatal care (PNC) and perception towards vaccine side effects did not show any significant association in the childhood vaccination status [30]. The

possible explanation for this disagreement could be due to differences in study design and sample size.

## Limitation of the study

Even though many variables relevant to this study were included to determine the independent predictors of incomplete vaccination, recall-related biases were introduced. These recall biases were due to the difficulty of mothers' who didn't have immunization cards to remember the immunization status of their child. Besides, our study introduced biases by respondents because of difficulty to guess the exact distance and time to the vaccination site. Despite the above limitations, our findings are important to understand factors associated with immunization completion among children in our study area.

## Conclusion

In conclusion, this study found that mothers or caretakers age being older ($\geq$ 35 years), with a negative attitude towards immunization, poor knowledge, lack of antenatal care service, delivery at home, lack of visit by a health professional within a year, and spending more than an hour reaching the vaccination site are all statistically significant and independent predictors of incomplete vaccination.

Therefore, the district health office and health professionals should work to create awareness about maternal health services, their importance, and need for complete immunization. Higher officials and health professionals should also prepare a plan and strategy to visit households at least once per month to trace the defaulters, and establish new vaccination sites to decrease long-distance travel.

## Supporting information

**S1 Dataset. Data using SPSS for incomplete immunization.**
(SAV)

## Acknowledgments

We would like to thank St. Paul's Hospital Millennium Medical College for giving chance, Amhara regional health bureau APHI and Worebabo district health office for their cooperation. Our appreciation also goes to health workers and officers working at all levels and directly involved in the data collection. Finally, my gratitude also goes to the study participants for providing us their time and giving us important information.

## Author Contributions

**Conceptualization:** Mesfin Yimer Abegaz, Shikur Mohammed Awol.

**Data curation:** Mesfin Yimer Abegaz, Shikur Mohammed Awol.

**Formal analysis:** Mesfin Yimer Abegaz, Shikur Mohammed Awol, Seid Legesse Hassen.

**Investigation:** Mesfin Yimer Abegaz, Awol Seid, Shikur Mohammed Awol.

**Methodology:** Mesfin Yimer Abegaz, Awol Seid, Shikur Mohammed Awol, Seid Legesse Hassen.

**Project administration:** Mesfin Yimer Abegaz.

**Resources:** Mesfin Yimer Abegaz.

**Software:** Mesfin Yimer Abegaz, Awol Seid, Shikur Mohammed Awol, Seid Legesse Hassen.

**Supervision:** Mesfin Yimer Abegaz, Awol Seid, Shikur Mohammed Awol, Seid Legesse Hassen.

**Validation:** Mesfin Yimer Abegaz, Awol Seid, Shikur Mohammed Awol, Seid Legesse Hassen.

**Visualization:** Mesfin Yimer Abegaz, Awol Seid, Shikur Mohammed Awol, Seid Legesse Hassen.

**Writing – original draft:** Mesfin Yimer Abegaz, Awol Seid.

**Writing – review & editing:** Mesfin Yimer Abegaz.

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
