## [Decision Letter · Decision Letter 0]

27 Mar 2023

PGPH-D-23-00136

Determinants of Incomplete Child Vaccination among Mothers of Children Aged 12–23 Months in Worebabo District of Amhara Region, North East Ethiopia: Unmatched Case-Control Study

Dear Dr. Abegaz,

Thank you for submitting your manuscript to PLOS Global Public Health. After careful consideration, we feel that it has merit but does not fully meet PLOS Global Public Health’s publication criteria as it currently stands. Therefore, we invite you to submit a revised version of the manuscript that addresses the points raised during the review process.

We look forward to receiving your revised manuscript.

Kind regards,

Collins Otieno Asweto, PhD

Academic Editor

Journal Requirements:

2. Please send a completed 'Competing Interests' statement, including any COIs declared by your co-authors. If you have no competing interests to declare, please state "The authors have declared that no competing interests exist". Otherwise please declare all competing interests beginning with the statement "I have read the journal's policy and the authors of this manuscript have the following competing interests:"

3. Please amend your online Financial Disclosure statement. If you did not receive any funding for this study, please simply state: “The authors received no specific funding for this work.”

4. Please provide separate figure files in .tif or .eps format only and remove any figures embedded in your manuscript file. Please also ensure that all files are under our size limit of 10MB.

5. Fig 1: please (a) provide a direct link to the base layer of the map (i.e., the country or region border shape) and ensure this is also included in the figure legend; and (b) provide a link to the terms of use / license information for the base layer image or shapefile. We cannot publish proprietary or copyrighted maps (e.g. Google Maps, Mapquest) and the terms of use for your map base layer must be compatible with our CC-BY 4.0 license. 

5. In the online submission form, you indicated that "All the datasets during and/or analyzed during the current study are available from the corresponding author on reasonable request". All PLOS journals now require all data underlying the findings described in their manuscript to be freely available to other researchers, either 1. In a public repository, 2. Within the manuscript itself, or 3. Uploaded as supplementary information.

**Comments to the Author**

1. Does this manuscript meet PLOS Global Public Health’s publication criteria? Is the manuscript technically sound, and do the data support the conclusions? The manuscript must describe methodologically and ethically rigorous research with conclusions that are appropriately drawn based on the data presented.

Reviewer #1: Yes

Reviewer #2: Yes

2. Has the statistical analysis been performed appropriately and rigorously?

Reviewer #1: Yes

Reviewer #2: Yes

3. Have the authors made all data underlying the findings in their manuscript fully available (please refer to the Data Availability Statement at the start of the manuscript PDF file)?

Reviewer #1: Yes

Reviewer #2: Yes

4. Is the manuscript presented in an intelligible fashion and written in standard English?

Reviewer #1: Yes

Reviewer #2: Yes

5. Review Comments to the Author

Reviewer #1: Please find the attached file

Title The title is very long. Please revise

Introduction Kindly reframe the sentence “as systematic identification of the factors”

Results Please revise the sentence “Also, the majority 224 (50.8%) of the mothers

Please revise the sentence “The majority 200 (45.4%) of mothers/caregivers of the

children in both study groups”

“Please revise the sentence Also, the majority 224 (50.8%) of the mothers”

While describing the results section, please discuss in the method section how you did you compute knowledge and then categorize it as good knowledge. The same for the positive attitude and positive perception

Health care utilization, please specify which model that have been followed to select predictors for the outcome variables. For example, health belief model

Please revise the sentence “Above two-thirds of the respondents, 374 (84.8%) from both groups

Please revise the sentence “Almost all mothers 119 (81 %) of the

cases and 278 (94.6%) controls who attended postnatal

Conclusion

Very long. Kindly revise to make concise and shorter sentences.

Reviewer #2: • Congratulations to the authors as their findings can influence strategies for closing the

immunization gaps in parts of Ethiopia.

• How was selection bias avoided between the cases and the controls in each kebele?

• What was the non-response rate in the study and how did it impact your results knowing that an assumption of

10% was factored into the sample size determination.

• Under introduction, in the second paragraph, change ‘’Homophiles’’ to ‘’Haemophilus influenza type B’’

• Under study design, line 2, change ‘’amon;; to ‘’among’’

• Under health care utilization, the second paragraph talks about ‘’ Above two-thirds of the respondents, 374

(84.8%) from both groups’’. Can this be rephrased since almost 85% is not two-thirds. Suggestion: ‘’Either you

use the percentage directly or you use more than 4 out 5’’

• Under discussion: paragraph two, line 3 should be changed from ‘’ Thus, children who born and get care from older

mothers’’ to ‘’ Thus, children born and cared for by older mothers’’ and on Line 11, change <=19 years to ≤19

years

6. PLOS authors have the option to publish the peer review history of their article (what does this mean?). If published, this will include your full peer review and any attached files.

**Do you want your identity to be public for this peer review?** For information about this choice, including consent withdrawal, please see our Privacy Policy.

Reviewer #1: **Yes: **Al-kubaisi, khalid Awad

Reviewer #2: No

---

## [Decision Letter · Decision Letter 1]

31 May 2023

Determinants of Incomplete Child Vaccination among Mothers of Children Aged 12–23 Months in Worebabo District, Ethiopia: Unmatched Case-Control Study

PGPH-D-23-00136R1

Dear Mesfin,

We are pleased to inform you that your manuscript 'Determinants of Incomplete Child Vaccination among Mothers of Children Aged 12–23 Months in Worebabo District, Ethiopia: Unmatched Case-Control Study' has been provisionally accepted for publication in PLOS Global Public Health.

Best regards,

Collins Otieno Asweto, PhD

Academic Editor

Reviewer Comments (if any, and for reference):

Reviewer's Responses to Questions

**Comments to the Author**

1. If the authors have adequately addressed your comments raised in a previous round of review and you feel that this manuscript is now acceptable for publication, you may indicate that here to bypass the “Comments to the Author” section, enter your conflict of interest statement in the “Confidential to Editor” section, and submit your "Accept" recommendation.

Reviewer #1: All comments have been addressed

Reviewer #2: All comments have been addressed

2. Does this manuscript meet PLOS Global Public Health’s publication criteria? Is the manuscript technically sound, and do the data support the conclusions? The manuscript must describe methodologically and ethically rigorous research with conclusions that are appropriately drawn based on the data presented.

Reviewer #1: Yes

Reviewer #2: Yes

3. Has the statistical analysis been performed appropriately and rigorously?

Reviewer #1: Yes

Reviewer #2: Yes

4. Have the authors made all data underlying the findings in their manuscript fully available (please refer to the Data Availability Statement at the start of the manuscript PDF file)?

Reviewer #1: Yes

Reviewer #2: Yes

5. Is the manuscript presented in an intelligible fashion and written in standard English?

Reviewer #1: Yes

Reviewer #2: Yes

6. Review Comments to the Author

Reviewer #1: All addressed

Reviewer #2: Congratulations to the research team.

7. PLOS authors have the option to publish the peer review history of their article (what does this mean?). If published, this will include your full peer review and any attached files.

**Do you want your identity to be public for this peer review?** For information about this choice, including consent withdrawal, please see our Privacy Policy.

Reviewer #1: No

Reviewer #2: **Yes: **Richard AMENYAH
